# A Review of CFD Analysis Methods for Personalized Ventilation (PV) in Indoor Built Environments

**Jiying Liu [1,2,\*]**, **Shengwei Zhu [2]**, **Moon Keun Kim [3,\*]** and **Jelena Srebric [2]**

1   School of Thermal Engineering, Shandong Jianzhu University, Jinan 250101, China
2   Department of Mechanical Engineering, University of Maryland, College Park, MD 20742, USA
3   Department of Architecture, Xi'an Jiaotong-Liverpool University, Suzhou 215123, China
\*   Correspondence: jxl83@sdjzu.edu.cn (J.L.); moon.kim@xjtlu.edu.cn (M.K.K.)

**Abstract:** Computational fluid dynamics (CFD) is an effective analysis method of personalized ventilation (PV) in indoor built environments. As an increasingly important supplement to experimental and theoretical methods, the quality of CFD simulations must be maintained through an adequately controlled numerical modeling process. CFD numerical data can explain PV performance in terms of inhaled air quality, occupants' thermal comfort, and building energy savings. Therefore, this paper presents state-of-the-art CFD analyses of PV systems in indoor built environments. The results emphasize the importance of accurate thermal boundary conditions for computational thermal manikins (CTMs) to properly analyze the heat exchange between human body and the microenvironment, including both convective and radiative heat exchange. CFD modeling performance is examined in terms of effectiveness of computational grids, convergence criteria, and validation methods. Additionally, indices of PV performance are suggested as system-performance evaluation criteria. A specific utilization of realistic PV air supply diffuser configurations remains a challenging task for further study. Overall, the adaptable airflow characteristics of a PV air supply provide an opportunity to achieve better thermal comfort with lower energy use based on CFD numerical analyses.

**Keywords:** computational fluid dynamics (CFD); personalized ventilation (PV); computational thermal manikin (CTM); inhaled air quality; thermal comfort; energy saving

## 1. Introduction

Conventional heating, ventilation, and air conditioning (HVAC) systems are used to maintain a uniform indoor environment in compliance with design standards. However, a certain population of occupants is more sensitive to local air movement, temperature, and relative humidity due to individual preferences as well as differences in clothing and activity levels. Therefore, thermal discomfort exists for a percentage of occupants even when the thermal environment complies with recommendations in the standards [1–3]. Moreover, symptoms of sick-building syndrome have resulted in increased attention to inhaled air quality based on individual needs and preferences [4]. Personalized ventilation (PV) aims to supply clean air directly to the breathing zone of each occupant to improve air quality. Several recent studies have introduced PV as a viable ventilation concept for office environments, and not just vehicles [3,5–7]. A paradigm shift from the design of indoor uniform environments to the design of individually controlled environments is advantageous even with respect to energy consumption in buildings. Several recent studies have found that individually controlled environments combined with a wider thermostat setpoint range for the background indoor environment result in greater thermal comfort and enable building energy savings [8–11].

The original PV studies for indoor built environments, such as offices and hospitals, (i) focused on delivering a small amount of personalized clean air directly to the breathing zone of each occupant [3,12]; (ii) improved thermal comfort [13,14]; and (iii) improved energy efficiency [11,15,16]. Numerous laboratory studies have demonstrated PV performance in different buildings and climate zones, including hot and humid climates [15,17,18] and cold climates [19]. PV is typically used in conjunction with total volume ventilation [10,20], such as mixing ventilation (MV) [21,22], displacement ventilation (DV) [16,23], and underfloor air distribution (UFAD) systems [24–26], or combined with liquid-based cooling systems, such as chilled ceilings [27,28]. PV is also considered as a standalone system with no assistive ventilation or cooling systems. For instance, a chair-based PV has been proposed for theatres, cinemas, and lecture halls [29]. The performance of PV on improving inhaled air quality, thermal comfort, and energy use has been widely studied.

In addition to experimental methods, computational fluid dynamics (CFD), as a simulation technique that can efficiently predict both spatial and temporal fields in the research of ventilation and indoor air science, has been adopted and studied [30–32]. Despite the dramatic increase in its application, CFD has not replaced experimental and theoretical analyses, but it has become increasingly important as a supplement to them [33–35]. In the PV field, a considerable number of studies have used CFD to analyze indoor air quality, thermal comfort, ventilation effectiveness, and energy saving in association with turbulence modeling, numerical approximations, and boundary conditions [36–39]. Compared to experimental methods, CFD can provide precise information regarding the distribution of flow and concentration fields in the whole simulation domain, rather than just targeted areas for data collection [40–42]. However, the quality of CFD simulation requires adequate grid generation, selection of numerical models, assessment of numerical errors, and control over other numerical parameters for proper verification and validation [43–46]. Therefore, as a system suitable for CFD analysis, PV requires a careful consideration of numerical parameters.

In recent years, several literature-review papers have been published in the PV research field. These reviews focused on different ventilation types supplemented by PV [47], personalized conditioning influencing thermal comfort and energy consumption [48], quantification of the ability of personal comfort systems to produce comfort [49], and advanced personal comfort systems in the workplace [50]. A number of review papers regarding CFD techniques in indoor built environments have been also published, including a CFD analysis procedure [51], quality control [52], application aspects and trends [53], turbulence models [54,55], verification and validation [56], pressure–velocity decoupling algorithms [57], CFD coupled with network modeling [58], large eddy simulation (LES), and Reynolds-averaged Navier–Stokes (RANS) approaches in building simulations [59], as well as reviews of CFD and indoor ventilation [30,33,60,61]. However, previous review studies have not conducted a PV review focusing on the CFD modeling approach, hence this paper targets this important knowledge base. The most relevant papers include a review of CFD studies to analyze and design the microenvironment around the human body [62], and a review of CFD studies of the thermal environment around a human body accounting for the PV system [63]. Nevertheless, with the increasingly fast development of the CFD technique and PV systems, a timely, critical literature review of CFD applications to PV systems is needed. Most importantly, there has been no comprehensive overview with respect to PV and its effect on the microenvironment around the human body.

This study reviews the existing knowledge base of CFD analyses focused on PV systems in indoor built environments. This review first provides an overview of CFD applications for PV in indoor built environments, starting with a brief introduction of PV research topics, CFD codes, PV types, and their development benefits from the perspective of CFD technical advancements. CFD setups of PV simulations are then reviewed, including turbulence models and model performance. This study specifically emphasizes the heat transfer around the human body. PV systems' parameters and their performance are analyzed, with a focus on PV air supply diffusers, PV air supply parameters, and evaluation indices. Through state-of-the-art analyses of current CFD applications in studying

PV systems, future CFD procedures and research interests are suggested to provide a preliminary guideline of PV analyses for HVAC engineers and researchers.

## 2. Research Methodology

A series of scientific databases—including ScienceDirect, SpringerLink, Taylor and Francis Online, Wiley Online Library, SAGE journals, and Web of Science, as well as the Google Scholar search engine—were employed for this scientific literature review. In combination with "CFD" or "numerical simulation", the following search keywords were used: "personalized ventilation", "task ambient conditioning", "air terminal diffuser", "task ambient air conditioning", "personalized conditioning", "personal thermal comfort", "personal cooling", and "personal heating". The searched articles were examined as they related to the aim of this review paper. Moreover, relevant literature known to the authors or reviewed through the above references was included. Additional studies that examine the capacity and accuracy of the CFD technique in simulating indoor built environments were also included.

A total 60 journal articles that studied PV's application in indoor built environments by CFD methods were reviewed in this study. Moreover, this review covered the reviews for CFD studies on indoor ventilation, which referred to many aspects of CFD simulation for indoor built environments, therefore were considered beneficial to this study [30,33,60,61]. Some important studies on CTM development were also included in the review, even they did not involve PV systems. However, those CFD studies on PV's application in vehicles, such as aircraft [64,65] and vehicle compartments [66,67], are excluded in this review as they are out of the scope.

## 3. Overview of Previous CFD Applications in PV in Indoor Built Environments

Compared to CFD studies of indoor ventilation, a comprehensive PV study using CFD simulations must account not only just for the geometric model, grid generation, turbulence model, boundary condition, numerical scheme, and post-processing, but also for the computational thermal manikin (CTM). Airflow interactions with the CTM surface significantly affect inhaled indoor air quality and thermal comfort induced by convective human body plume or forced jet flow [68]. Therefore, CTM in PV studies is reviewed. Sixty journal articles focus on PV and CFD. Table 1 shows an overview of CFD analysis of PV, with the following categories: (1) author(s) and publication year, (2) PV type, (3) CFD code, (4) turbulence model, (5) radiation model, (6) CTM surface boundary condition, (7) near wall treatment, and (8) target parameters. The primary overview covers the yearly publication distribution, CFD codes, research topics, and PV types with background HVAC systems, to provide a development status of PV studies using CFD.

**Table 1.** CFD analyses of PV in indoor built environments.

| Author & Year | PV with BV | CFD Code | Turb. | Rad. | BC (no.) | NWT | Target Parameters |
|---|---|---|---|---|---|---|---|
| Gao and Niu 2004 [36] | PV with DV | NS | SKE | NO | Tsk(1) | SWF | AT, AV, CHTC, CO2, PER, PPD, PUE |
| Gao and Niu 2005 [69] | PV with DV | Fluent | RNG | NS | Tsk(1) | EWT | AT, AV, CHTC, CO2, PER |
| Gao et al. 2006 [37] | DMPV or CBPV with DV, RMP | Fluent | RNG | YES | Tsk(16) | EWT | ACE, AT, AV, AVV, CHTC, LTC, LTS, OTC, OTS, PER |
| Nielsen et al. 2007 [70] | PV with MV | NS | k-ε | NS | q(1) | NS | ACH, AV, AVV, PEI |
| Gao et al. 2007 [71] | PV with DV or MV | Fluent | SKE | YES | Tsk(16) | NS | AT, AV, AVV, Cinh, LTC, LTS, OTC, OTS, PER |
| Zhao and Guan 2007 [72] | PV | STACH-3 | ZEQ | NS | q(1) | No | AT, AV, AVV, PC |
| Yang and Sekhar 2008 [73] | PV or CMPV with MV | Fluent | SKE | NO | NO | NS | AV, AVV, FAP, PEE |
| Russo et al. 2009 [74] | PV with UFAD | ANSYS Fluent | RKE | NS | Tsk(20) | EWT | AT, AV, PEE, TI |
| Dygert et al. 2009 [75] | PV with UFAD | Fluent | SKE, RKE, RNG, SKW, SST | NO | Tsk(1) | EWT | AQI, AV, AT |
| Russo and Ezzat Khalifa 2010 [76] | PV | NS | NS | NS | Tsk(1) | EWT | IF, PEE, Sct |
| Russo and Khalifa 2010 [77] | PV | Fluent | RKE | NS | Tsk(1) | EWT | IF, Ozone |
| Conceição et al. 2010 [78] | DMPV | NS | k-ε | NS | Tsk(15) | NS | AT, AV, DR, PMV |
| Tham and Pantelic 2010 [79] | DMPV and DF with MV | Fluent | SKE | NO | Tsk(26) | EWT | AV, CHF, PEE, Teq |
| He et al. 2011 [80] | RMP with DV, MV, UFAD | Fluent | RNG | S2S | q(1) | SWF | AV, AVV, IF, PC, Rc, TGC |
| Zhai and Metzger 2011 [81] | PV with MV | PHOENICS | RNG | NO | NS | NS | AGE, ES, PMV |
| Mazej and Butala 2011 [82] | MSDP with UFAD | ANSYS Fluent | RNG | S2S | Tsk(16) | EWT | AV, AT, AVV, CHTC, CO2, PEE, RHTC, RIE, Teq, TGC |
| Adamu et al. 2011 [83] | NPV | PHOENICS | RNG | NS | Q(1) | NS | AGE, AT, ATT, AV, CRE, LACH, MACE |
| Ishiguro et al. 2011 [84] | CMPV | NS | NS | NO | Q(1) | NS | AV, AT, AVV, PMV, Qc |
| Russo and Khalifa 2011 [85] | PV | NS | NS | NS | Tsk(1) | NS | AV, AT, AQI, IF, TGC |
| Li et al. 2012 [86] | CBPV or DMPV with MV or DV | Fluent | RNG | NS | q(1) | SWF | ACE, AT, AV, AVV, IF, PC |
| Kanaan et al. 2012 [87] | PV with DV | Airpak | SKE | YES | Q(1) | NS | AT, AV, AVV, CO2, PEE |
| Shen et al. 2013 [38] | RMP or VDG with DV, MV or UFAD | ANSYS Fluent | SKE | S2S | q(1) | SWF | AT, AV, AVV, MFP, TGC, VE |
| Makhoul et al. 2013 [88] | CMPV and DF | ANSYS Fluent | RKE | NS | Tsk(11) | EWT | AV, AT, AVV, CO2, ES, LTC, OTC, PEE |
| Makhoul et al. 2013 [89] | CMPV | ANSYS Fluent | RKE | NS | Tsk(11) | EWT | LTC, LTS, OTC, OTS, PEE, Qc |
| Makhoul et al. 2013 [90] | CMPV | ANSYS Fluent | RKE | NS | Tsk(-) | NS | AT, IF, PC, PVV |
| Yang et al. 2013 [91] | DMPV or VDG with MV | ANSYS Fluent | SKE | NS | Tsk(1) | EWT | CO2, IF, PEE |

**Table 1.** *Cont.*

| Author & Year | PV with BV | CFD Code | Turb. | Rad. | BC (no.) | NWT | Target Parameters |
|---|---|---|---|---|---|---|---|
| Cheong and Huang 2013 [92] | RMP or DMPV with DV | Fluent | RNG | NS | Tsk(26) | NS | AV, AT, AVV, PEIc |
| Makhoul et al. 2013 [93] | CMPV | ANSYS Fluent | RKE | NS | Tsk(11) | EWT | AV, AT, $CO_2$, PEE |
| Yang and Sekhar 2014 [94] | CMPV with MV | Fluent | SKE | NO | NO | NS | AV, AVV, PEE, PAP, TGC |
| Russo and Ezzat Khalifa 2014 [95] | PV with UFAD | Fluent | RKE | NO | Tsk(1) | EWT | Ozone, IF |
| Kong et al. 2015 [39] | PV with UFAD | STAR-CCM+ | RKE | NS | Tsk(20) | TLWT | AT, AV, AVV, PEE, Teq, TGC |
| Naumov et al. 2015 [96] | PV | ANSYS | NS | NS | Tsk(-) | NS | AV, AT, EXT, ES |
| Shao and Li 2015[97] | PV | Airpak | ZEQ | NS | Q(1) | NO | DPSA, TACS, TAIC, TASA |
| Antoun et al. 2016 [98] | CMPV | ANSYS Fluent | RKE | S2S | Tsk(11) | EWT | AV, LTC, LTS, OTS, OTC, Qc, RHF, TI, Tsk |
| Abou Hweij et al. 2016 [99] | CF with DV | ANSYS Fluent | SKE | NS | Tsk(11) | EWT | AT, AV, LTC, LTS, OTC, OTS, Tsk, PEE |
| El-Fil et al. 2016 [100] | CMPV and CF | ANSYF Fluent | RKE | NO | Tsk(11) | EWT | AT, AV, $CO_2$, LTC, LTS, OTC, OTS, PEE, Qc |
| Zhu et al. 2016 [101] | WCPV with MV | STAR-CD | RNG | NO | q(1)&Tsk(1) | SWF | AT, AV, DR, PEE, SVE3, SVE3*, SVE4 |
| Conceição et al. 2016 [102] | CMPV | NS | RNG | YES | Tsk(25) | NS | AQN, AT, AV, $CO_2$, ADI, DR, PPD, Tsk |
| Habchi et al. 2016 [103] | CMPV and CF or DF | ANSYS Fluent | RKE | NO | Tsk(11) | EWT | AT, AV, $CO_2$, DFr, IF, LTC, OTC, PC, PEE, Qc |
| Habchi et al. 2016 [104] | CMPV and DF | ANSYS Fluent | RKE | NO | q(20) | EWT | AT, AV, DFr, IF, PC, Qc |
| Mao et al. 2016 [105] | BTAC | ANSYS Fluent | SST | S2S | Tsk(16) | NS | AV, AT, AVV, RH, Tsk |
| Taheri et al. 2016 [106] | PV with DV | NS | NS | NS | NS | NS | AT, AV, $CO_2$, PMV, TSV |
| Zhu et al. 2017 [107] | RPAC with MV | Fluent | RNG | YES | Tsk(17) | EWT | AT, AV, AVV, CE, DR, OTS, Tsk, ΔPMV |
| Conceição et al. 2017 [108] | DMPV | NS | RNG | NS | Tsk(15) | NS | AV, AT, $CO_2$, DR, MRT, PMV, PPD, Tclo, Tsk, TI |
| Mao et al. 2017 [109] | BTAC | ANSYS Fluent | SST | S2S | Tsk(1) | NS | AT, AV, PMV, Qc |
| Mao et al. 2017 [110] | BTAC | ANSYS Fluent | SST | S2S | Tsk(1) | NS | AT, AV, PMV, Qc |
| Ahmed et al. 2017 [111] | LEVO with DV | ANSYS Fluent | RNG | DO | q(1) | EWT | AV, AT, PC, PMV, PPD, Qc, VATD |
| Ahmed and Gao 2017 [112] | LEVO with DV | ANSYS Fluent | RNG | DO | q(1) | EWT | AV, AT, DR, PC, Qc, VTAD |
| Al Assaad et al. 2017 [113] | PV with MV | ANSYS Fluent | RNG | NS | Tsk(11) | EWT | AT, AV, $CO_2$, FRF, OTC, PEE, Qc, Tsk |
| Du et al. 2017 [114] | BTAC with radiant panel | ANSYS Fluent | SST | S2S | Tsk(1) | NS | AT, AV, EUC, DR, PMV, To |
| Kong et al. 2017 [115] | RMP or DMPV with MV | STAR-CCM+ | SKE | YES | Tsk(20) | TLWT | AT, AV, CHF, CHTC, To, Tsk |
| Sun et al. 2017 [116] | DMPV with cooling ceiling | Airpak | SKE | DO | Q(1) | NS | AT, DR, PMV, PPD, To, ES |
| Alotaibi et al. 2018 [117] | CMPV and DF or CF | ANSYS Fluent | RKE | NS | q(1) | NS | AV, AT, CI, PC, Qc, ΔC |
| Alsaad and Voelker 2018 [118] | DPV with DV | ANSYS Fluent | SKE, RKE, RNG | S2S | Tsk(16) | EWT | AT, AV, $CO_2$, LTC, LTS, OTC, OTS, PEIc, |
| Al Assaad et al. 2018 [119] | PV with cooling ceiling | ANSYS Fluent | RNG | S2S | Tsk(10) | EWT | AT, $CO_2$, OTC, OTS, PEE, Qc, TI |
| Al Assaad et al. 2018 [120] | DMPV with MV | ANSYS Fluent | RNG | NS | q(1) | EWT | AT, AV, $CO_2$, DFr, FRF, IF, PC |

**Table 1.** *Cont.*

| Author & Year | PV with BV | CFD Code | Turb. | Rad. | BC (no.) | NWT | Target Parameters |
|---|---|---|---|---|---|---|---|
| Conceição et al. 2018 [121] | DMPV | NS | RNG | NS | Tsk(25) | NS | ADI, AQN, AT, AV, CO2, DR, PPD |
| Gao et al. 2018 [122] | TPV | NS | RSM | NS | NS | NS | AV, AT, Qc, Ts, Tr |
| Rahmat et al. 2018 [123] | DF and DMPV with UFAD | Airpak | ZEQ | S2S | Q(1) | NO | AT, AV, PMV, PPD, Qc, VATD |
| Sekhar and Zheng 2018 [124] | PV with ACB | ANSYS Fluent | RKE | NS | q(1) | SWF | AT, AV, AVV, PPD, PMV, Qc, |

**Abbreviations:** PV type with background ventilation type (PV with BV), new wall treatment (NWT), radiation model (Rad.), turbulence model (Turb.), CTM boundary condition type with segment number (BC (no.)); PV with BV: bed-based task/ambient air conditioning (BTAC), chair fan (CF), chair based personalized ventilation (CBPV), ceiling mounted personalized ventilation (CMPV), desk mounted personalized ventilation (DMPV), desk fan (DF), ductless personalized ventilation (DPV), local exhaust ventilation for offices (LEVO), movable slot diffuser plate (MSDP), natural personalized ventilation (NPV), robotic personal air conditioning (RPAC), round movable panel (RMP), targeted personalized ventilation (TPV), vertical desk grille (VDG), wide-cover personalized ventilation (WCPV); Background HVAC system type: displacement ventilation (DV), mixing ventilation (MV), underfloor air distribution (UFAD), local fan-induced active chilled beam air conditioning system (ACB); Turbulence modes: low Reynolds number k-ε (LRNKE), realizable k-ε (RKE), Reynolds stress models (RSM), renormalization group k-ε (RNG), SST k-ω (SST), standard k-ε (SKE), zero-equation turbulence model (ZEQ); CFD codes: not specified (NS); Near wall treatment: enhanced wall treatment (EWT); standard wall function (SWF), two-Layer all y+ wall treatment (TLWT); Radiation model: discrete ordinates radiation model (DO), surface-to-surface radiation model (S2S); Target parameters: (i) local ventilation and air quality: air change rate (h$^{-1}$) (ACH), carbon dioxide concentration (ppm) (CO2), concentration asymmetry (-) ($\Delta$C), deposited fraction (DFr), inhaled contaminant concentration (ppm) (Cinh), intake fraction (IF), particle concentration (kg/m$^3$) (PC), re-inhaled exposure index (-) (RIE), scale for ventilation efficiency 3 (-) (SVE3), new scale for ventilation efficiency 3 (-) (SVE3*), scale for ventilation efficiency 4 (-) (SVE4), tracer gas concentration (ppm) (TGC), personal exposure index (-) (PEI), personal exposure effectiveness (-) (PEE); (ii) Thermal comfort: change of predicted mean vote (-) ($\Delta$PMV), local thermal comfort (-) (LTC), local thermal sensation (-) (LTS), overall thermal comfort (-) (OTC); overall thermal sensation (-) (OTS), predicted percentage dissatisfied (%) (PPD), draught risk (%) (DR), predicted mean vote (-) (PMV), CTM-based equivalent temperature (°C) (Teq), vertical air temperature difference at the head and foot level (°C) (VATD), mean radiant temperature (°C) (MRT), operative temperature (°C) (To); (iii) Energy saving: cooling efficiency (-) (CE), energy utilization coefficient (-) (EUC), calculated energy consumption (W) (Qc); Others: age of air (s) (AGE), air distribution index (-) (ADI), air quality number (-) (AQN), air temperature (°C) (AT), air velocity (m/s) (AV), air velocity vector (-) (AVV), air turnover time (s) (ATT), concentration uniformity index (-) (Rc), convective heat flux (W/m$^2$) (CHF), confinement index (%) (CI), convective heat transfer coefficient (W/m$^2$K) (CHTC), contaminant removal efficiency (CRE), cloth surface temperature (°C) (Tclo), derivation of the index of difference potential by supply air (-) (DPSA), excess temperature (°C) (EXT), flow rate frequency (Hz) (FRF), fresh air percentage (%) (FAP), local air change index (%) (LACH), mean air exchange efficiency (%) (MACE), mass fraction of pollutants (%) (MFP), ozone concentration (ppm) (Ozone), particle velocity vector (-) (PVV), radiative heat flux (W/m$^2$) (RHF), radiative heat transfer coefficient (W/m$^2$K) (RHTC), relative humidity (%) (RH), skin surface temperature (°C) (Tsk), transient accessibility of contaminant source (-) (TACS), transient accessibility of initial condition (-) (TAIC), transient accessibility of supply air (-) (TASA), turbulent Schmidt number (-) (Sct).

### 3.1. Yearly Publication Distribution

Published in 2004 [36], the first article in the scope of PV and CFD studied the microenvironment around the human body resulting from a PV system operation. Figure 1a shows the yearly distribution of CFD studies of PV systems, which are increasingly popular. It is found that the number of CFD studies of PV in the last three years already exceeds 45% of all of the studies.

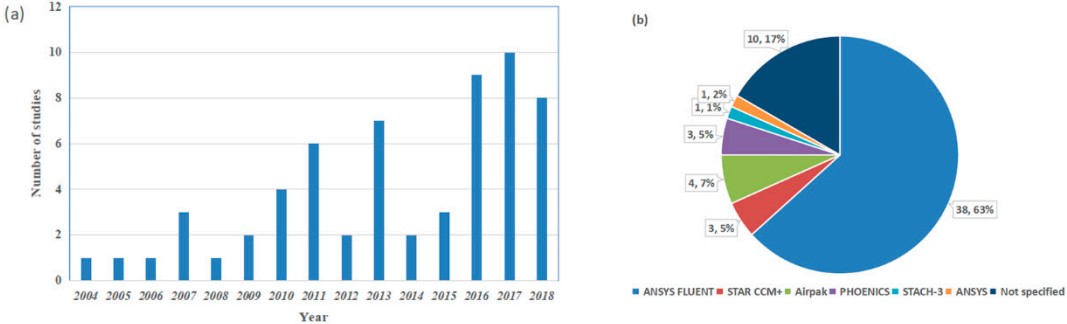

**Figure 1.** Distribution of reviewed journal papers on PV studies using CFD. (**a**) Year of publication; (**b**) CFD code.

### 3.2. CFD Codes

Figure 1b shows the CFD code distribution of CFD studies of PV systems. The figure shows that of all of the publications reviewed in this study, 38 of 60 articles (63%) employed ANSYS Fluent software (including the early version, called Fluent). Other commonly used programs are STAR-CCM+, Airpak, PHOENICS, STACH-3, and ANSYS. A study compared ANSYS Fluent and STAR-CCM+, which are currently the most popular, in predicting indoor air parameter distributions [125]. The study pointed out that the two programs produced almost the same results with similar computing effort, and emphasized that ANSYS Fluent was more user-friendly, owing to its user-defined functions. Authors do not promote any commercial CFD package, and encourage diverse application of CFD in PV studies, including the open-source OpenFOAM, which has powerful grid capabilities and a wide range of applications [126]. Note that Airpak is no longer updated, but it has been used in recent studies [123].

### 3.3. Research Topics of PV Studies

Research topics of PV using CFD are categorized into three major types: (i) local ventilation and inhaled air quality, (ii) thermal comfort, and (iii) energy savings, based on the target parameters summarized in Table 1. Local ventilation and inhaled air quality are evaluated based on the air change rate ($h^{-1}$) (ACH), carbon dioxide concentration (ppm) ($CO_2$), concentration asymmetry (-) ($\Delta C$), deposited fraction (DFr), inhaled contaminant concentration (ppm) ($C_{inh}$), intake fraction (-) (IF), particle concentration ($kg/m^3$) (PC), pollutant exposure reduction (%) (PER), ventilation effectiveness (-) (ES), personal exposure effectiveness (-) (PEE), scale for ventilation efficiency 3 (-) (SVE3), new scale for ventilation efficiency 3 (-) (SVE3*), scale for ventilation efficiency 4 (-) (SVE4), re-inhaled exposure index (-) (RIE), and tracer gas concentration (ppm) (TGC). Thermal comfort studies use local thermal comfort (-) (LTC), local thermal sensation (-) (LTS), overall thermal comfort (-) (OTC), overall thermal sensation (-) (OTS), predicted mean vote (-) (PMV), predicted percentage dissatisfied (%) (PPD), draft risk (%) (DR), CTM-based equivalent temperature (°C) ($T_{eq}$), operative temperature (°C) (Top), vertical air temperature difference at the head and foot level (°C) (VATD), and mean radiant temperature (°C) (MRT). Energy saving studies evaluate the cooling efficiency (-) (CE), energy utilization coefficient (-) (EUC), and calculated energy consumption (W) ($Q_c$).

It is important to note that the target parameters used in Table 1 are not the recommended evaluation indices, but instead are the main parameters used in literature focused on validation,

comparison, or performance evaluation PV systems with CFD. Being consistent with the original aim of studying PV systems, the research topic category of local ventilation and air quality constitutes almost 80% (46 of 60) of the reviewed studies. Thermal comfort and energy saving were investigated by 28 studies and 18 studies, respectively. Additionally, in the last three years, the three major types of research topic—local ventilation and air quality, thermal comfort, and energy saving—have the proportions of 20:18:16 in a total of 27 studies, which shows almost equal importance. This distribution coincides with two of the current challenges in building science: (i) the growing need for comfort improvements and (ii) increased concern for energy consumption [48]. The commonly used evaluation indices will be described in the next section.

### 3.4. PV Types with Background HVAC Systems

A comprehensive study presented five air terminal devices (ATDs), namely, a movable panel (MP), computer monitor panel (CMP), vertical desk grill (VDG), horizontal desk grill (HDG), and personal environment module (PEM) [6]. Another study redesigned a round movable panel (RMP) [127], used in several review studies [38,80,92]. The commonly used PV type also includes ceiling-mounted personalized ventilation (CMPV) [88,89], chair-based personalized ventilation (CBPV) [29,37], desk-mounted personalized ventilation (DMPV) [120], desk fan (DF) and/or chair fan (CF) [103,104], and bed-based task/ambient air conditioning (BTAC) [105,109,110]. Figure 2 shows the commonly used PV types. Note that several PV types included in relatively few articles are not presented in this figure and can be found in the references. These less popular PV types include ductless personalized ventilation (DPV) [118], local exhaust ventilation for offices (LEVO) [111,112], movable slot diffuser plate (MSDP) [82], natural personalized ventilation (NPV) [83], robotic personal air conditioning (RPAC) [107], targeted personalized ventilation (TPV) [122], and wide-cover personalized ventilation (WCPV) [101].

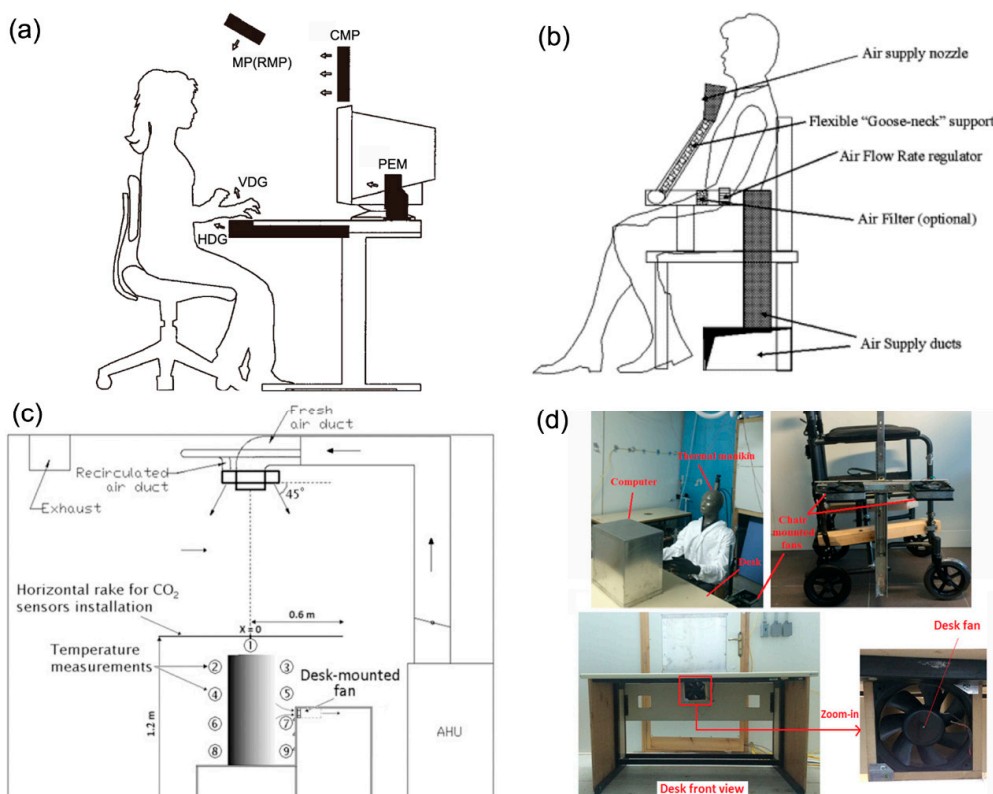

**Figure 2.** Commonly used PV types. (**a**) ATDs (modified from [6]), (**b**) CBPV [29,37], (**c**) CMPV [88,89] and DMPV [120], (**d**) DF and CF [103,104].

Depending on individual preferences, especially in large offices with several workstations, occupants may adjust their PV system supply flow rate or temperature, or even switch it off. Therefore, the background environment must maintain an acceptable comfort level. The background HVAC system can be used in combination with PV, resulting in a whole space with non-uniform velocity and temperature fields and unevenly distributed pollutants [3]. Background HVAC systems are classified into two types: (i) background ventilation and (ii) radiant cooling/heating. The former is a convection-dominated system, i.e., total volume system ventilation [10], including MV (14 studies), DV (14 studies), and UFAD (8 studies). The latter is radiation-dominated and uses radiant panels [114,128] or a chilled ceiling [116,119]. It should be noted that some PV systems recirculate air in addition to using outside air by utilizing a low-mixing PV nozzle integrated with a four-sided slot diffuser [89,93], as shown in Figure 2c. The supply PV nozzle creates a jet that can penetrate the thermal plume around the human body and reach an occupant's breathing zone with the necessary amount of outside air to maintain acceptable inhaled air quality.

## 4. Modeling Computational Thermal Manikin

In reviewed PV studies, modeling of CTMs plays a significant role in determining the distributions of airflow temperature, velocity, and pollutant concentrations. These distributions directly affect inhaled air quality, occupant thermal comfort, and energy consumption. The personal microenvironment is highly influenced by the convective heat transfer between the body surface and ambient environment, radiant heat transfer between the body surface and wall surface, and buoyancy-driven thermal plume around the body. Therefore, it is important to identify best CFD practices for implementation of realistic CTMs. This review explores related topics, including turbulence models, heat transfer around CTMs, and modeling performance.

### 4.1. Turbulence Models

PV studies primarily employ RANS methods. The three most commonly used RANS turbulence models are the renormalization group k-ε turbulence model (RNG) [129] (20 of 60 studies), realizable k-ε turbulence model (RKE) [130] (16 of 60 studies), and standard k-ε turbulence model (SKE) [131] (13 of 60 studies). Shear stress transport k-ω (SST) [132] was used in four studies, zero-equation turbulence model (ZEQ) [133] in three studies, and Reynolds stress models (RSMs) [134] in one study. Table 1 presents additional details on the turbulence models used in studies of PV systems.

For RANS-based models, selecting a proper turbulence model is a prerequisite for conducting performance evaluations of PV systems. In general, a PV system represents a mixed convection heat-transfer mode, including human body buoyancy-driven thermal plumes, natural convection induced by background ventilation, and forced convection due to PV nozzle supply airflow. Several studies have evaluated the performance of turbulence models. A study simulated a low-mixing co-flow nozzle directing a PV fresh-air jet toward the breathing zone of a seated occupant using SKE, RKE, RNG, the k-omega model (SKW), and SST turbulence models [75]. The results showed that the k-ε family of turbulence models presented similar results and agreed well with experimental data, while the SKW and SST turbulence models performed poorly at predicting jet deflection/distortion and jet spreading. For instance, for average convective heat flux, the k-ε family of turbulence models overestimated the values by up to 1%, while the k-ω family of turbulence models over-predicted the values by up to 4%. Similarly, the results of SKE and RNG are in good agreement with experimental data, while performance of the k-ω group models does not agree well with experiments in forced and mixed convection cases [135]. Another study validated a conventional round nozzle delivering fresh air directly to an occupant's face using STK, RKE, and RNG [118]. RKE achieved the best results based on the air temperature, ventilation effectiveness index, and air velocity between the PV outlet and the occupant's face. Another study evaluated the performance of the SKE, RNG, and SST models for predicting vertical profiles of air temperature and air velocity distribution around a sleeping occupant using bed-based task/ambient air conditioning; SST achieved a better agreement than the other models

with measured data [105]. A study compared SKE, RKE, SST, and RSM for predicting the air velocity around the human head using a personalized square diffuser, and pointed out that RSM performed better than the other models [122].

Besides the reviewed comparisons for different turbulence models, most studies followed recommendations outlined in CFD program manuals and/or guidelines for CFD applications in indoor/enclosure environments, to choose appropriate turbulence models among a wide range of popular ones. For instance, among the RANS models, RNG was extensively used due to its stable performance in simulations of both forced and mixed convection around the human body [107], accuracy and good predictability of the jet spreading rate and the behavior of recirculated air [113], and better overall performance in terms of accuracy, computing efficiency, and robustness [81]. RKE is also popular because of its high accuracy in predicting the spreading rate of planar and round jets, and its superior predictions of flows involving separation and recirculation in indoor environments [88,103,117]. Moreover, the RKE model is effective for both high and low Reynolds number turbulent flows, and can therefore produce good-quality results even when the flow is not fully turbulent [74]. Finally, SKE is widely used for CFD simulations of PV airflows in enclosed environments because it can simulate convective heat transfer of buoyancy-driven airflows with reasonably acceptable results [36,79], despite its many deficiencies, such as inaccurate predictions of turbulence kinetic energy and adverse pressure gradient flows [71].

High Reynolds number turbulence models are almost always applied in PV studies due to the development of enhanced wall treatment (EWT, 25 of 54 studies) in ANSYS fluent and/or two-layer y+ wall treatment (TLWT, 2 of 54 studies) in STAR CCM+. The EWT approach uses a near-wall modeling method that combines a two-layer model with enhanced wall functions. If the near-wall mesh is fine enough (typically y+≈1), then EWT will automatically resolve the viscous sublayer. Additional details on the grid distribution adjacent to human body surfaces are provided in the following section. Furthermore, low Reynolds k-ε turbulence models (LRN) were typically used to account for convective heat transfer with the locally-fine grid adjacent to the body surface before applicable wall functions were adopted [62,136,137]. LRN can predict the transport process throughout the boundary layer with a large number of computational cells, in contrast to the standard k-ε model, which is valid only for fully developed turbulent flows [52]. However, LRN models converge with difficulty due to the complicated geometry associated with extremely fine grids near the human body [63], so their popularity is waning.

It should be noted that few studies utilize the modified v2f-dav model, detached eddy simulation model (DES), or the more robust but computationally demanding LES, which have been found to perform better for force convection and mixed convection in ventilated spaces [54,55]. A study concluded that v2f-dav and RNG showed the best overall performance compared to other models in terms of accuracy, computing efficiency, and robustness [138]. The v2f-dav model is not commonly applied in indoor environments, probably because it is unavailable in commercial CFD software. Another assessment of turbulence models for transitional flows in an enclosed environment [139], demonstrated that the RNG and RSM models could achieve a good overall performance to characterizing a flow before it became too complex, especially when these two models were used to simulate mixed convection in a room with a heated box. DES with sub-grid-scale turbulence kinetic energy and LES still has the most accurate and stable performance in enclosed environments [54,55]. LES provided the most detailed flow features and accuracy, but used significant computing time [61].

Although no single turbulence model can optimally and economically handle all flow elements—e.g., jet flow, momentum-driven flow, stratified flow, and buoyancy-driven flow [30]—there is still a great necessity to utilize more accurate turbulence models or predict the human body microenvironment characterized by a complex flow characteristic. In fact, LES has been applied to predict the microenvironment around a manikin, mostly using simplified geometry [140–142]. With increasing computer power, LES may gain popularity for PV studies. Further study is required to

investigate the use of more advanced models, to improve the balance between CFD simulation accuracy and required computational effort.

### 4.2. Heat Transfer around the Human Body

Accurate predictions of heat transfer around the human body have direct implications on accuracy of predicted thermal comfort and inhaled air quality with PV systems. When introducing CTMs in CFD modeling of PV systems, it is important to consider CTM's geometry, heat exchange models, boundary conditions, and thermoregulation models.

### 4.2.1. Geometry of CTMs

There are no relevant standards for the sizes, shapes, and postures of CTMs, and their usage mostly depends on the purposes of research studies. CTMs used a surface area of 1.57 $m^2$ [36,92], 1.59 $m^2$ [69], or 1.8 $m^2$ [115], and a simple shape, e.g., rectangular, or exact-shape, as well as sleeping, seated, or standing postures. Most studies used complex CTM geometry, the exception being a rectangular person used to assess the performance of a CMPV system in regard to inhaled air quality influenced by a moving person [94]. This is mainly due to a complex flow field and the extensive computation required. Note that in Airpak software, the CTM geometry is predefined as a rectangular shape of some limited body parts and can only be changed according to the posture or height/length, because its library function stores or retrieves groups of objects in a parts library [143].

Computational requirements are mostly driven by the geometric complexity of the human body, resulting in different levels of findings adjacent to the body surface. Theoretically, the more realistic the CTM geometry the more accurate the results. Although existing literature has not examined the effect of CTM geometry simplification on PV performance, a few studies have qualified the influence of geometric complexity on an indoor environment, e.g., the shape and size of CTM [144]. One study pointed out that a detailed manikin shape can provide more accurate predictions (4–10%) at some locations, especially those close to the body [141]. Another study revealed that although human body simplification only affected airflow field prediction of the thermal plume regions, it could increase the predictive error of contaminant transport in the whole computational domain [145]. Furthermore, a study proposed a surface smoothing approach that can decrease the number of grids on the CTM surface to reduce the mesh independence effect, improving computational efficiency while obtaining a reliable evaluation. This illustrates the importance of keeping the key body features when simplifying human body [146]. Therefore, it might be possible to utilize more accurate representation of critical body surfaces, such as the head and face, while simplifying the other human body surfaces.

### 4.2.2. Heat Exchange between Human Body and Microenvironment

The current CFD modeling technique only accounts for heat exchange between the human body and the microenvironment, instead of within the human body, i.e., CTMs do not have the thermoregulation functions that can respond to the microenvironment as human bodies do [147]. The CTMs are mainly used for heat dissipation from the human body, including heat loss by radiation, sensible heat loss by convection, and latent heat loss by convection [62]. For instance, the human body releases its metabolic heat production to the surrounding environment, 49.3 W by convection, 65.1 W by radiation, 41.3 W by evaporation, and 14.8 W by respiration [136]. Moreover, a total of 115 W for a seated person with very light work was generated, including 70 W of sensible heat loss due to convection and radiation, and 45 W of latent heat loss induced by evaporation and respiration, representing the amount of heat released to the surrounding environment to achieve thermal balance [115].

As the PV flow involves mixed convection in most articles, the buoyancy effect should be accounted for. The commonly used method to include thermal buoyancy utilizes the Boussinesq approximation for incompressible flow, and an incompressible ideal gas model. The Boussinesq approximation is applied in indoor airflows as long as the density variations are small or $\beta(T - T_O) \ll 1$ [135], which cannot be combined with the species transport equation in the RANS model [148]. The study found

that buoyancy tends to suppress turbulence at a stable stratification but promotes it at an unstable stratification [149]. Turbulence production or dissipation by buoyancy can be incorporated when using SKE, RNG, SKE, SST (using a user-defined function), and RSM models. Therefore, the full buoyancy-effects option may be selected to consider the effect of buoyancy on the turbulent dissipation rate ($\varepsilon$), since it is not well understood [148]. The predictive accuracy of the two models above is evaluated in mixed ventilation except in the case of a high-temperature heat source [150].

For radiation models in 20 of 60 studies, the surface-to-surface (S2S) and discrete ordinate radiation (DO) models were mostly applied in the reviewed papers in ANSYS Fluent [148], while S2S and the participating radiation model were selected in the Star CCM+ program [151]. We do not intend to explain the radiation models, whose details can be found in program tutorials. Instead, we provide the following analysis for setting boundary conditions once the radiation is considered. A review evaluated three thermal radiation approaches when simulating normal indoor thermal flows in terms of predictive accuracy and computational time in the ANSYS CFX program [152], demonstrating the importance of including thermal radiation in the CFD model [153].

### 4.2.3. Boundary Conditions

The thermal boundary condition for CTMs representing the heat source can be modeled in three ways: constant surface temperature (35 of 60 studies), constant surface heat flux (12 of 60 studies), and total heat power output through the volume called a constant volumetric heat generation for the human body (5 of 60 studies). The last is popularly used in the Airpak and PHOENICS programs, and is employed in the ANSYS Fluent program, although the study did not consider the PV case [154].

In general, the CTM surface temperature/heat flux can be selected from the measured data [74,79,92,110] or estimated from the literature and/or experience for simplification [38,80]. For simulations disregarding radiation, the convective heat flux per unit area was determined for the CTM surface boundary condition, e.g., as 39 W/m$^2$ [117,120]. When accounting for radiation effects, some articles still utilized surface heat flux [38,80,111,112]. As the studies simplified the boundary condition setup with a uniform heat flux value, it was necessary to determine the convection-to-radiation ratio (C:R) to properly calculate the convective heat flux. Many values of C:R were found in the reviewed literature, such as 43:57 for a seated person with a surface area of 1.69 m$^2$ [136], 40:60 for a seated person with a surface area of 1.59 m$^2$ and sensible heat loss of 89 W [52], 42:58 for a seated person with 1.45 m$^2$ surface area in a uniform radiation environment [155,156], 40:60 for a standing person with 1.48 m$^2$ surface area and 76 W [56], 30:70 for a standing person with 1.46 m$^2$ surface area [144], 38:62 for a seated person with 1.48 m$^2$ surface area [157], and 60:40 for a seated person using CFD radiation [158]. Note that those variable values emphasize the importance of modeling radiative heat flux between human body and the microenvironment together with the convective heat fluxes.

Note that it is expensive to calculate radiative heat transfer from a human body to room walls by using a surface-to-surface radiation model, because of the large number of segment surfaces accounting for the irregular human body shape [63,159]. Therefore, radiation heat transfer is commonly avoided by assuming that a certain percentage (usually half) of CTMs' heat loss was caused by convection. The heat flux boundary condition is then used by calculating the ratio of convective heat loss at CTM surface area. If the research is only aimed to globally capture the effect of the thermal plume, the convection-to-radiation ratio might be sufficient [160]. Review found that 11 studies utilized one uniform heat flux value. However, research has shown that it was erroneous to neglect radiation modeling when heat flux boundary conditions were used [157,161]. Using heat flux cannot represent the exact heat-loss portions of convection, radiation, and evaporation, because of their nonlinear relationships, and their local distributions in the segments. Therefore, it is not realistic to only consider convective heat transfer around the human body in an indoor environment, especially when radiation plays a significant role.

A total of 12 studies adopted one uniform temperature value for whole body surfaces because they were focused on PV system properties that were not significantly affected by non-uniform heat transfer

around the human body. A study pointed out that when a predefined temperature was used for the body surface, the heat transfer was independent of thermal radiation, which might be disregarded in the simulations [52]. To avoid neglecting about half of the heat loss, using a temperature boundary condition is crucial for proper specification of CTM heat exchange between surfaces and the ambient environment [115]. In addition, even when the PV is used to provide clean air to inhalation, instead of providing local cooling or heating, it is also important to take into consideration the non-uniform temperature distribution over a human body, because of its impact on thermal plume around the human body. Before being inhaled, personalized air needs to penetrate thermal plume first. Therefore, it is relatively correct to define the actual surface temperatures in a realistic environment, and then the effects of radiation can be included without a radiation model [157]. However, obtaining a detailed description of the CTM surface temperature distribution is difficult in an experiment. The commonly used method is to obtain the average temperature for the major segment surface and set it as the constant temperature boundary condition [160]. In addition, coupling CFD with a thermoregulation model was used to define the thermal boundary condition, which will be presented in next section. The representative surfaces, including the indoor floor and wall temperature, are best measured in advance. This can be a drawback because the surface temperature should first be determined. Overall, it is more accurate to set non-uniform thermal boundary conditions when seeking local evaluation indices, e.g., local thermal comfort and specified segment temperatures. There is no standard for the number of segments required for indoor environment studies. The number of segments may be consistent with the manikin number used in the experiments. Studies of segment numbers are listed in Table 1.

Although it was demonstrated that breathing of a sedentary manikin had a very small impact on the occupant thermal plume [162], discrepancies in the healthy manikin's exposure to exhaled contaminated air still existed between simulations and experiments, which could possibly be attributed to disuse of the transient inhalation and exhalation [91]. Therefore, some studies (10 of 60) considered the breathing process of CTMs, although most only took into account the steady-state inhalation [36,37,69,71] or exhalation process [38,86,87], or both [101], instead of a completely transient breathing process [82,85]. The constant inhaled flow rate was about 0.14 L/s [37,71], and the exhaled flow rate using the nose or mouth ranged from 0.02 L/s [87] to 0.14 L/s [37,91,103], with an exhaled air temperature of 34 °C [38,87,101] or 35 °C [86], depending on the activity level. Note that owing to the research topics that are mainly predicting particle transport and assessing ventilation effectiveness, most studies presented that particle-generation characteristics were considered as horizontal generation steady rate with mass flow rate of $5 \times 10^{-5}$ kg/s from normal breathing of the infected person [103,104].

For the transient breathing process, a study described the inhalation and exhalation process as a sinusoidal curve function consisting of inhalation (2.5 s), exhalation (2.5 s), and a pause (0.5 s), with an amplitude flow rate of 0.3 L/s [82]. This study demonstrated the usefulness of CFD in investigating the influence of the breathing process on inhaled air quality, and the necessity of proper breathing simulation to understand the complex airflow interactions in the breathing zone. Another study compared steady and transient breathing methods, and recommended the use of steady-state inhalation as the breathing method instead of exhalation, as it does not increase the complexity of the simulation compared to unsteady methods [85]. In addition, it is necessary to model the transient breathing process when studying re-inhaled air and the transport of exhaled contaminants, although exhalation concentration was found to have little effect on the inhaled concentration because of the strong momentum of the PV air supply to carry the pollutant away from the breathing zone. Given the limited number of studies addressing this issue, more studies are suggested to evaluate the effect of a steady or transient breathing process. Note that the detailed respiration cycle and its flow rate were reviewed by Gao and Niu [63].

### 4.2.4. Coupling of CFD and Thermoregulation Models

CTMs do not include thermoregulation models to define the heat output of the body by default. However, non-uniform microenvironments and transient environments are typically encountered

in buildings and in situations where a person's job involves exposure to high- or low-temperature surfaces or sources [159]. Therefore, knowledge of human thermal response to non-uniform and transient environments is important to accurately provide thermal boundary conditions to a CFD solver using thermoregulatory models. The coupled process includes the data exchange with respect to body segment heat fluxes and temperatures, and detailed local microenvironment parameters [163]. An early PV study coupled CFD and the thermoregulation model to investigate PV in regard to inhaled air quality and local thermal comfort [37]. This model consisted of 16 body segments and considered heat and moisture transfer as well as convection, conduction, and radiation [164,165]. Similarly, another study examined the thermal adaptive effect by changing the human body's orientation or position relative to spot airflow by means of CFD coupled with Fanger's neutral model [166]. A CFD study for PV integrated with a transient bio-heat model included the computation of segmental skin temperatures, overall thermal sensation, and overall thermal comfort [113]. In addition to PV studies, coupling CFD and a human-body thermoregulation model has been extensively employed in common studies in indoor environments [147,155,156,161,167,168].

There are already several comprehensive reviews of thermoregulation models [163,169–171]. The accuracy of transferred parameters of the environment around the human body in PV studies is vital to the accuracy of CFD simulation results. The quality of coupling is highly dependent upon CTM geometry, heat transfer around CTMs, clothing, and computational resources [161]. In addition, the coupling process might hinder the analyses of PV system performance if convergence is difficult to achieve [172]. Thus, the primary task for the application of coupled CFDs with thermoregulation models is still to ensure the accuracy and efficiency of CTM simulations.

*4.3. Modeling Performance*

Any CFD study, including studies of PV systems, needs attention to modeling performance focused on computational grids, convergence criteria, and validation.

4.3.1. Computational Grids

The computational grid has attracted more attention with respect to quality control [52], verification [56], evaluation errors [158], and sensitivity [173] because the computational results depend greatly on the grid quality, particularly due to complex CTM geometry [63]. This section outlines recent progress in the use of computational grids for PV studies, and suggests grid-independence analyses using evaluation indices.

Table 2 summarizes the grids in the computational domain and around CTMs for a few studies. Due to the complex human body geometry, hybrid grids—including tetrahedral, hexahedral, and prismatic grids—were commonly adopted in the simulation domain. Figure 3 shows different types of hybrid grids created in a room and around a human body. The prismatic grids were used in the human body boundary layer. In early PV studies, in addition to the boundary layer (prismatic grid), hexahedral combined with tetrahedral grids were typically used for the ambient domain, owing to the utilization of Gambit [174]. Tetrahedral and prismatic grids were mainly used to study complex CTM shapes and indoor configurations [175]. Honeycomb and hexahedral grids were used for the majority of PV studies in Star CCM+ [39,115].

**Table 2.** Grid distribution in the computational domain and around CTMs

| Authors and Year | CDD | TCN(LCN) | TCNP | FLH(NBL) | MSCS | y+ | MSA | Grid Type |
|---|---|---|---|---|---|---|---|---|
| Gao and Niu 2004 [36] | 2.6 × 2.2 × 2.7 | 1.81(1.59) | 0.12 | | | | 1.57 | Hex/Tet |
| Gao and Niu 2005 [69] | 2.6 × 2.2 × 2.7 | 1.75(1.06) | 0.11 | | | most < 1 | 1.59 | Hex/Tet |
| Russo, Dang et al. 2009 [74] | 2.0 × 2.6 × 2.5 | 4.2 | 0.32 | 1.5(4) | <12 | <3 | | Hex/Tet |
| Tham and Pantelic 2010 [79] | 5.55 × 3.7 × 2.6 | 2.64(2.08) | 0.05 | 2(15) | | <1 | | Hex/Tet |
| Shen, Gao et al. 2013 [38] | 5.4 × 4.8 × 2.6 | 1.08 | 0.02 | | | | | |
| Makhoul, Ghali et al. 2013 [88] | 1.7 × 3.4 × 2.8(half) | 1.2 | 0.07 | 1.5(4) | 10 | 0.8–4 | 1.78 | |
| Makhoul, Ghali et al. 2013 [89] | 1.7 × 3.4 × 2.8(half) | 1.21 | 0.07 | 1.5(4) | 20 | 0.8–4 | 1.78 | |
| Makhoul, Ghali et al. 2013 [90] | 1.7 × 3.4 × 2.6(half) | 1.21 | 0.08 | | | - | - | |
| Makhoul, Ghali et al. 2013 [93] | 1.7 × 3.4 × 2.6(half) | 1.21 | 0.08 | 1.5 | | 0.8–4 | 1.78 | |
| Cheong and Huang 2013 [92] | 6.6 × 3.7 × 2.7 | 2.71(2.16) | 0.04 | | | most < 1 | 1.57 | Hex/Tet |
| Yang, Sekhar et al. 2013 [91] | 6.6 × 3.7 × 2.6 | 4.61 | 0.07 | 2.2(2) | | 0.5–4 | | |
| Kong, Zhang et al. 2015 [39] | 1.9 × 1.8 × 1.7 | 0.34 | 0.06 | (5) | 50 | 1.58 | | Hon |
| Antoun, Ghaddar et al. 2016 [98] | 1.7 × 2.7 × 2.6 (half) | 1.42 | 0.12 | (3) | | 0.8–4 | | |
| Abou Hweij, Ghaddar et al. 2016 [99] | 2.5 × 2.75 × 2.8 | 3.52 | 0.18 | | | 0.8–4 | | |
| El-Fil, Ghaddar et al. 2016 [100] | 3.4 × 3.4 × 2.8 | 2.83 | 0.09 | 1.8(3) | 20 | 0.9–4.5 | | |
| Al Assaad, Ghali et al. 2017 [113] | 2.8 × 2.75 × 2.5 | 1.06 | 0.06 | 1.5 | | | | |
| Ahmed, Gao et al. 2017 [111] | 4.0 × 2.7 × 3.0 | 2.75 | 0.08 | 1.5 | | 0.7–4.5 | | |
| Ahmed and Gao 2017 [112] | 4.0 × 2.7 × 3.0 | 2.75 | 0.08 | 1.5 | | 0.7–4.5 | | |
| Du, Chan et al. 2017 [114] | 3.7 × 2.6 × 3.29 | 2.44(1.45) | 0.08 | 0.4 | | | | Hex/Tet |
| Kong, Dang et al. 2017 [115] | 4.8 × 3.66 × 3.05 | 5.0 | 0.09 | 1(10) | 2.5-10 | <0.8 | 1.8 | Hon |
| Alsaad and Voelker 2018 [118] | 3.0 × 3.0 × 2.4 | 5.77 | 0.27 | | | most < 1 | | |
| Al Assaad, Habchi et al. 2018 [120] | 3.4 × 3.4 × 2.8 | 6.5 | 0.20 | | 15 | 0.8–4 | | Tet |
| Al Assaad, Ghali et al. 2018 [119] | 2.5 × 2.75 × 2.8 | 2.11 | 0.11 | | 15 | 0.8–4 | | Tet |
| Alotaibi, Chakroun et al. 2018 [117] | 3.4 × 3.4 × 2.6 | 2.84 | 0.09 | | | | | |
| Gao, Wang et al. 2018 [122] | 3 × 1.5 × 3 (half) | 1.6 | 0.12 | | | | | |

**Abbreviations:** computational domain dimension (m × m × m) (CDD), total cells number (-) (TCN), local zone cell number around the human body (-) (LCN), first layer height (mm) (FLH), number of inflated prismatic layers (-) (NBL), manikin surface cell size (mm) (MSCS), non-dimensional wall distance in manikin surface (-) (y+), manikin surface area (m$^2$) (MSA), grid type (GY), hexahedral grid (Hex), tetrahedral grid (Tet), honeycomb grid (Hon), total cell number per volume unit (TCNP).

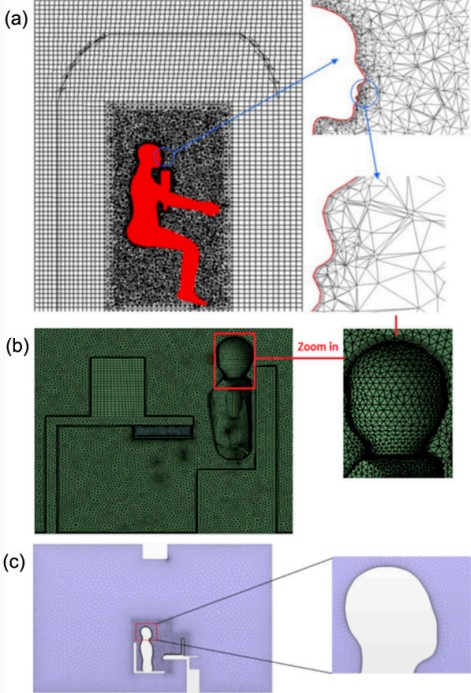

**Figure 3.** Grid distribution in room and around the human body. (**a**) Hybrid grids (prismatic, tetrahedral, and hexahedral) [69], (**b**) Hybrid grids (tetrahedral and prismatic) [117], (**c**) Hybrid grids (honeycomb and hexahedral) [115].

The total grid number reached 5.77 million and 6.5 million, representing 0.27 million grids and 0.20 million grids per cubic meter, respectively, in two recent studies [118,120]. However, two early studies used only 1.81 million and 1.75 million grids, resulting in 0.12 million/m$^3$ and 0.11 million/m$^3$, respectively [36,69]. Future studies will employ larger grid numbers in line with increasing computational capacity. However, CFD studies may become more inefficient than experimental studies, even with the development of computer technology, especially when considering dynamic simulation scenarios. Therefore, a balance between efficiency and accuracy of simulations must be considered.

The range for the initial inflated prismatic height adjacent to the human body varied from 0.4 mm [114] to 2.2 mm [91], and y+ values on the majority of the body's surface were lower than 4.5 in all cases, which is consistent with non-PV studies [56,158]. Most studies utilized a maximum CTM surface grid size of up to 20 mm. Based on this literature review, a prismatic layer number of at least 3 is suggested, with the expansion ratio under 1.2, which results in sufficiently good grid quality represented by equi-volume skewness within the recommended value of CFD solvers [56,160]. Moreover, y+ varies with the location, and the maximum value occurs mostly in the breathing zone close to the body, so the local grid refinement is required to place the first grid point within the viscous sublayer [113].

Verification of CFD includes the grid convergence index (GCI), which is used to compare discrete solutions at different grid spacing [44,176]. Note that GCI for validation of CFD was suggested for the grid refinement analysis [52]. However, this approach received little attention in PV studies. This is probably due to the difficulty in selecting variables and locations for comparisons, so a new normalized root mean square error index (RMSE*, less than 10%) developed from the GCI index, and a numerical viscosity analysis method may be applied for a grid independence analysis in indoor environments [173]. Grid independence analysis must be sufficiently considered, instead of simply comparing absolute errors or relative errors for limited variables in an ambient environment or on the body surface.

### 4.3.2. Convergence Criteria

The default convergence criteria in CFD programs are not always convincible when the residual root mean square error value is below the criterion (typically $10^{-4}$ or $10^{-5}$) [52]. Therefore, besides the condition when the residual values reach an acceptable level for a steady-state condition, two additional assessment indices are required to ensure convergence. One is that the overall imbalance for the specific variables in the domain should be less than 1%, and the other is that the monitored point for the values of interest should reach a steady solution.

In PV studies, the convergence criterion was reached not only when the scaled residuals met the requirement, but also when velocity magnitude at specific points in the thermal plume and breathing zone had stabilized [74,75]. A net heat flux of less than 1% of the total heat gained in the domain was used as the convergence criterion [113,120]. Another study examined the sensitivity of iterative convergence in an indoor environment, and indicated that iterative convergence criteria have a large impact on the results of the age of air [154]. It was suggested that convergence can be reached when the net heat flux imbalance of the domain boundary is less than 1%. In addition, the criteria were employed when the convective and sensible heat losses from the human body were stabilized with the iteration [107]. Another study judged numerical convergence using scaled residuals, net heat flux of total heat gain, and stabilization of $CO_2$ concentration in the breathing zone [93]. Overall, PV studies have clearly taken into account the overall convergence criteria.

### 4.3.3. Validation

The fundamental strategy of validation is to assess how accurately the computational results compare with the experimental data, with quantified error and uncertainty estimates for both [45]. Unfortunately, no sufficient estimates of uncertainty or error were provided for CTMs until a detailed validation study was conducted in an indoor environment [56,160]. Through the validation, the aspects of grid convergence, turbulence model, and radiation model were identified and qualified, accounting for errors associated with both the computational model and experiment. Practically, it is not possible for every study to pursue a comprehensive validation. Note that the majority of studies selected a computational model based on research experience or references. In 2003, a study established a benchmark test case including a CTM in an MV or DV system for validation of CFD models [177]. A subsequent study provided another benchmark test case involving PV combined with a general ventilation system in a room from the experiment [178,179]. The latter test case has been applied in several publications [39,74,76,77,85]. Therefore, it is a useful database to compare simulated results with a PV benchmark test to validate new programs, turbulence models, grid dependency, and numerical schemes [30].

## 5. PV Air Supply, Thermal Comfort, and Energy Savings

The most important design considerations for a PV system include air supply characterized by air supply diffusers and air supply parameters. The performance of PV air supply defines how successful such a system is in delivering desired local thermal comfort and inhaled air quality with potential energy savings for the central HVAC unit.

### 5.1. PV Air Supply Diffusers

An experimental study validated that the air terminal device within three different layouts of blinds with the same diffuser size and effective area had a significant influence on the occupant's thermal sensations [180]. Therefore, correct modeling of a PV air supply diffuser is an important prerequisite to accurately predict the microenvironment around a human body. The commonly used models involve the simplified boundary condition method, box-method, prescribed velocity method, and momentum method [30]. Note that the air supply diffusers for the background ventilation system are not discussed in detail, as extensive studies have investigated diffuser boundary conditions

without PV diffusers [181–183]. This literature review reveals that the majority of PV studies used the momentum method, which is specified as a momentum source term in the conservation equation and is calculated from the mass flow rate and effective diffuser area. This is probably because the relatively small air supply opening area is incapable of modeling the detailed diffuser geometry because it requires a large number of grids. Another explanation is that the momentum method is more effective and saves time in setting diffuser boundary conditions. It is noticeable that few studies have paid specific attention to the effect of real diffuser geometry, which has mostly occurred in general CFD cases of indoor microenvironments in recent years, e.g., inserted lobes [184], blade of louver [185], and swirl blades [186].

An exception for PV studies utilized the measured data in a PV and UFAD air supply diffuser as a boundary condition, as shown in Figure 4, and reported that the inlet velocities have non-uniform distributions [39]. For instance, the largest velocity occurred in the central two slots of the PV diffuser, especially for large flow rates. Thus, given that PV flow directly affects the occupant's body within a short distance, the shape, configuration, or inner components in air supply diffusers—e.g., guide vanes, perforated plates, sound dampers, and curved surfaces—can have a positive influence on thermal comfort [180,187] and ventilation effectiveness [188]. Future studies using PV air supply diffusers with their actual geometries are necessary to evaluate their effect on the microenvironment around the human body [30].

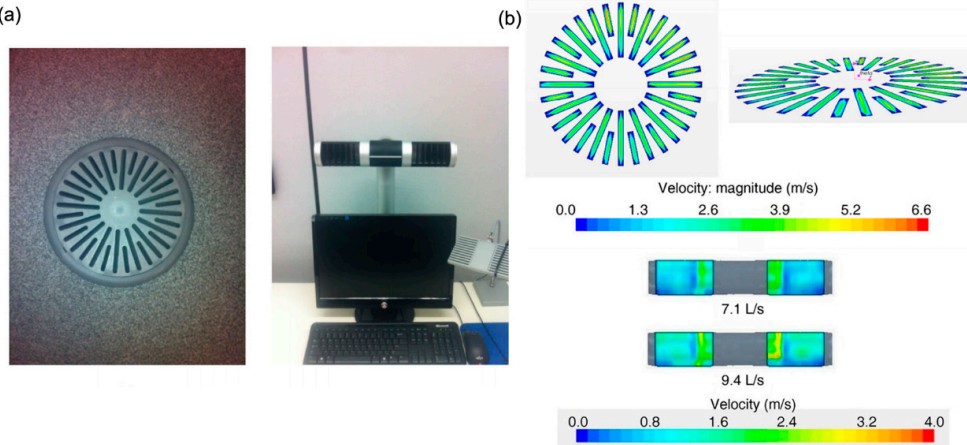

**Figure 4.** (**a**) UFAD and PV configuration view, and (**b**) velocity distribution [39].

### 5.2. PV Air Supply Parameters

To evaluate the performance of PV systems, CFD studies require PV air supply parameters, such as air velocity or flow rate, temperature, air humidity, and trace gas concentration. This review aims to provide an overview of CFD applications in PV studies, suggestions for CFD simulation setups, and further research interests. Thus, air supply parameters for typical PV with a background ventilation system are not reviewed in detail. Instead, we compare commonly used parameters, such as flow rate, air temperature, and turbulence intensity.

Table 3 summarizes air supply parameters in a typical PV and background ventilation system. It shows that the majority of studies selected an airflow rate from 2.4 to 20 L/s, and an air temperature from 16 to 24 °C to meet occupant needs. For desk fans or chair fans, the supply boundary conditions can be set as fan boundary conditions with predefined pressure changes according to the flow rates [99,100], or inlet boundary conditions with an assumed temperature of ambient air [79]. Background ventilation systems typically have flow rates of 14 to 60 L/s per person and air temperatures from 16 to 24 °C. The air turbulence intensity mostly varied from 2.5% to 10% for PV and 2.5% to 15% for background ventilation. The distances between PV supply openings to an occupant's face/head ranged from 0.2 to 0.6 m for an air terminal device around a desk, or about 1.4 m for PV mounted in the ceiling.

**Table 3.** Summary of air supply parameters in a typical PV and background ventilation system.

| Authors and Year | PV System | | | Background Ventilation System | | | Distance (m) |
|---|---|---|---|---|---|---|---|
| | Flow Rate (L/s) | Tin (°C) | Ti (%) | Flow Rate (L/s) | Tin (°C) | Ti (%) | |
| Gao and Niu 2004 [36] | PV0~3 | 20 | 0.5 | DV14 | 22 | 5 | <0.1 |
| Kong, Zhang et al. 2015 [39] | PV7.1, 9.4 | 21.7 | | UFAD48.8 | 24.3 | | |
| Gao and Niu 2005 [69] | PV0.8~1.6 | 18~24 | 5,10,20 | DV14 | 22 | 10 | <0.1 |
| Gao, Zhang et al. 2007 [71] | PV10~20 | 17~21 | 20 | MV/DV/31~51 | 17~21 | MV30, DV15 | |
| Russo, Dang et al. 2009 [74] | Pri0.6~4.8, Se1.7~13.4 | Pri23.5 | Pri/Se1.7~10 | UFAD0.7~16.6 | 21 | | 0.4 |
| Russo and Khalifa 2010 [76,77] | PV2.4, Sec6.7 | 21 | 1.7 | UFAD10.1 | 21 | 1.7 | 0.4 |
| He, Niu et al. 2011 [80] | RMP7~15 | 20 | | DV/MV/UFAD25~40 | 20 | | 0.586 |
| Mazej and Butala 2011 [82] | PV5,10 | 20,23 | 5 | DV120 | 20,23,26 | 10 | |
| Russo and Khalifa 2011 [85] | PV2.4, Sec6.7 | 21 | 1.7 | UFAD10.1 | 21 | 1.7 | 0.4 |
| Li, Niu et al. 2012 [86] | CBPV/DMPV0.8~6.5 | 18 | | DV/MV22~28 | 20 | | |
| Kanaan, Ghaddar et al. 2012 [87] | PV4~10 | 18~22 | | DV116 | 18 | | 0.3~0.5 |
| Cheong and Huang 2013 [92] | PV5, 10, 15, 20 | 19~23 | 5, 10 | DV60 | 23 | 10 | |
| Makhoul, Ghali et al. 2013 [89,93] | Pri5~10, Sec10~20 | Pri16~24, Sec26~28 | | Dif10~20, 40~57 | 16 | 2.7 | 1.49 |
| Shen, Gao et al. 2013 [38] | RMP/VDG7,15 | 17 | 10 | DV/MV/UFAD10~23 | 17 | DV5, MV/UFAD15 | |
| Yang, Sekhar et al. 2013 [91] | PV4, 8 | 23 | 10 | MV(-) | 23 | 10 | 0.15~0.55 |
| Antoun, Ghaddar et al. 2016 [98] | Pri8.5, Sec(-) | 16, Sec(-) | 3 | Dif50, 55, 60 | 16, 18, 19 | 2 | 1.6 |
| El-Fil, Ghaddar et al. 2016 [100] | CMPV8.5, CF2.5 | 16, CF(-) | 2.5 | Dif50 | 16 | 2.5 | 1.4 |
| Habchi, Chakroun et al. 2016 [103] | Pri8.5,11, CF10, DF10 | 16 | | Dif35 | 16 | | 1.4 |
| Habchi, Ghali et al. 2016 [104] | Pri10, DF5,10,15 | 16 | | Dif35 | 16 | | 1.4 m |
| Zhu, Cai et al. 2016 [101] | WCPV2.5,5 | 18,19 | 10 | AC24 | 24 | 10 | |
| Al Assaad, Ghali et al. 2017 [113] | PV3.5,5,7.5 | 22 | | MV63 | 20 | | 0.4 |
| Kong, Dang et al. 2017 [115] | PV8~38 | 21~25 | 10 | MV75 | 27.8, 28.3 | 10 | 0.2~0.61 |
| Zhu, Dalgo et al. 2017 [107] | RPAC11.8~59 | 19~24.6 | 10 | MV52.5 | 26 | 10 | |
| Alsaad and Voelker 2018 [118] | DPV5,6.5 | | | DV16,24,43 | 19,22 | | 0.4 |
| Alotaibi, Chakroun et al. 2018 [117] | CMPV8.5, DF10, CF10 | 16 | 2.5 | Dif35 | 16 | 2.5 | 1.4 |
| Al Assaad, Ghali et al. 2018 [119] | PV3.5,5,7.5 | 22 | | MV63 | 15 | | 0.4 |

**Abbreviations:** personalized ventilation (PV), ceiling mounted personalized ventilation (CMPV), desk mounted personalized ventilation (DMPV), chair based personalized ventilation (CBPV), desk fan (DF), chair fan (CF), round movable panel (RMP), vertical desk grille (VDG), ductless personalized ventilation (DPV), robotic personal air conditioning (RPAC), wide-cover personalized ventilation (WCPV), displacement ventilation (DV), mixing ventilation (MV), underfloor air distribution (UFAD), primary nozzle (Pri), secondary nozzle (Sec), diffuser in the CMPV (Dif).

Figure 5 shows two special PV nozzles, including a reduced-mixing personal ventilation jet [74] and low-mixing coaxial nozzle mounted in the ceiling (CMPV) [89,93]. For example, CMPV has two nozzles, the primary nozzle delivering outside air at a flow rate ranging from 5 to 11 L/s, and the secondary nozzle supplying recirculated air at a flow rate ranging from 10 to 20 L/s. The review for PV and background ventilation air supply parameters are given as further references. Not all of the studies provide the minimum information that should be included in CFD-related papers, as required [43].

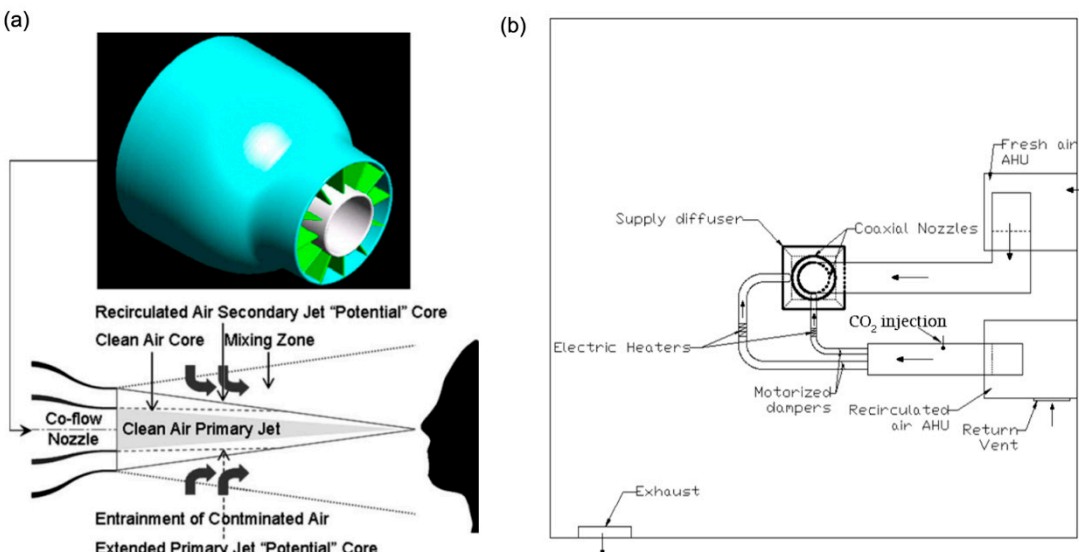

**Figure 5.** (**a**) Low mixing co-flow nozzle [179] and (**b**) low-mixing ceiling-mounted nozzle [88].

As a powerful tool, CFD has a significant advantage in capturing flow characteristics and particle trajectories. However, the majority of studies, especially in the early stage of development, only accounted for the steady-state condition. For the indoor environment, a study pointed out that airflow cannot be expected to be in steady-state condition, and this can only be predicted for a given flow with time-dependent equations [30]. Another study found that air-movement parameters, such as turbulence intensity and fluctuation frequency, can significantly affect human thermal perception, and suggested that dynamic airflows can achieve stronger cooling effects and better thermal comfort evaluation without negatively impacting occupants' work performance [189]. A few well-known unsteady studies utilized CFD to assess the performance of PV under the influence of a moving person [73,94], the breathing process [82,85], and sinusoidal airflow with different fluctuation frequencies [113,119,120]. One aspect of dynamic airflow is characterized by perceived air movement with a range of fluctuation frequency, e.g., 0.5 and 1.0 Hz [190], where the range of associated amplitudes and mean velocities is determined by minimum and maximum jet flows [113]. Suitable air velocities enable the penetration of the thermal plume around the human body to reach the breathing zone. Furthermore, an acceptable temperature range to deliver cold/warm air to the human body with ambient air is guaranteed to maintain better thermal comfort.

The fluctuation frequency in PV studies using CFD has only gained attention in the last two years, while its effect on human thermal response was extensively examined in experiments starting in the early 21st century [191–193]. Compared to constant flow, a sinusoidal airflow with the right frequency can have a greater cooling effect on the same average velocity [194]. An average flow rate combined with a specific frequency can guarantee good breathable air quality and comfort, providing a good compromise between thermal comfort and air quality, and better protection from direct contamination [120]. Therefore, to some extent, to use average flow rate solely in steady state is not an adequate simulation method. Moreover, other aspects of dynamic airflow involve the frequent change of an occupant's posture and position, transport of simulated natural wind, and coughing and sneezing behavior. All of these aspects have an important impact on PV performance, which can be

fulfilled by CFD technique. Therefore, further CFD work is required to account for the use of PV in more realistic environments.

## 5.3. Performance Evaluation Indices

Several evaluation criteria emerged to assess performance of PV systems, including inhaled air quality, thermal comfort, and energy savings.

### 5.3.1. Inhaled Air Quality

PV considers the occupants' health, comfort, and productivity around a human microenvironment. Its performance can be assessed based on occupant-related performance criteria, e.g., improvement of inhaled air quality, improvement of thermal comfort, and protection from and minimization of contaminant transmission [3]. Evaluation indices for ventilation effectiveness and inhaled air quality have been introduced in the literature [6]. However, these criteria were developed for experimental studies, and recent studies employ new indices specifically developed for CFD studies [88,103,113,119]. Table 4 summarizes commonly used evaluation indices for ventilation effectiveness and inhaled air quality. It is found that, due to the powerful post-processing functions in CFD programs, relevant parameters in specific locations/regions can be monitored. For instance, a study calculated the ventilation effectiveness based on tracer gas concentrations at the breathing zone that was defined as a small sphere of 1 cm radius at 2.5 cm from the occupant's nose [88,103]. Moreover, the cross-infection level can be assessed by monitoring the number of particles deposited in the vicinity of an exposed person and the number of particles generated by an infected person using deposited fraction (DFr) [103,120]. A few other evaluation indices have been applied in PV studies using CFD, e.g., new scale for ventilation efficiency 3 (SVE3*) and scale for ventilation efficiency 4 (SVE4) [101], age of air (AGE) [81], and concentration asymmetry ($\Delta$C) [117]. Note that SVE3 and SVE4 respectively represent the mean age and residual life of air, without considering where air is from and where it is exhausted, while SVE3* and SVE4* respectively represent the mean age and residual life of air from a specific inlet or outlet [195]. They have been used in few PV studies; details can be found in Table 1.

**Table 4.** Evaluation indices for ventilation effectiveness and inhaled air quality

| Index | Definition | Explanation for Parameters | Purpose of the Evaluation Index | Typical References |
|---|---|---|---|---|
| EV1 | $\varepsilon_{EV} = \frac{C_e - C_S}{C_j - C_S}$ | $C_e$ tracer gas concentration at the exhaust duct<br>$C_S$ tracer gas concentration at the supply duct<br>$C_j$ tracer gas concentration at the simulated point (e.g., mouth of the exposure manikin) | Assess the air distribution efficiency in rooms and around a human body | [80,118] |
| EV2 | $\varepsilon_{EV2} = \frac{C_p - C_S}{C_e - C_S}$ | $C_p$ pollutant concentration at simulated point (inhalation zone)<br>$C_e$ pollutant concentration at the exhaust opening<br>$C_S$ pollutant concentration at the supply air | Evaluate the effect of PV on inhaled air quality and trace gas pollutant transport characteristics around a polluting occupant | [38] |
| EV3 | $\varepsilon_{EV3} = \frac{C_r - C_b}{C_r - C_f}$ | $C_r$ tracer gas concentration of the recirculated air or at outlet<br>$C_f$ tracer gas concentration of the fresh air<br>$C_b$ tracer gas concentration at the breathing zone defined as a small sphere of 1 cm radius, at 2.5 cm from the occupant nose | Assess the mixing level of recirculated air and delivered fresh air supplied by second nozzle and diffuser and primary nozzle, respectively, using the tracer gas | [88,103,113,119] |
| PEI | $\varepsilon_{PEI} = \frac{C_j - C_S}{C_{sf} - C_S}$ | $C_j$ pollutant concentration at the simulated point<br>$C_S$ pollutant concentration at the supply air<br>$C_{sf}$ pollutant concentration on the surface of pollutant source | Assess the extent of the measuring location affected by the polluting source | [92] |
| PER | $\varepsilon_{PER} = \frac{C_L - C_a}{C_f - C_a}$ | $C_L$ tracer gas concentration of inhaled air<br>$C_f$ tracer gas concentration of personalized air<br>$C_a$ tracer gas concentration of ambient air. | Expressed as the fraction of personalized air in inhaled air | [36,37,69,71] |
| PEE | $\varepsilon_{PEE} = \frac{C_{I,0} - C_I}{C_{I,0} - C_{PV}}$ | $C_{I,0}$ concentration of pollution in the inhaled air without PV<br>$C_I$ concentration of pollution in the inhaled air<br>$C_{PV}$ concentration of pollution in personalized air | Estimate personal exposure effectiveness for PV in inhaled air, $C_{PV} = 0$ means that PV provides clean air with no pollutants | [82,87,91,101] |
| PEEc | $\varepsilon_{PEEc} = \frac{C_{R,SF6} - C_{I,SF6}}{C_{R,SF6} - C_{PV,SF6}}$ | $C_{RS6}$ SF6 concentration in the re-circulated air duct<br>$C_{PV,SF6}$ SF6 concentration of the supplying fresh air<br>$C_{I,SF6}$ SF6 concentration at the breathing zone | Estimate personal exposure effectiveness for SF6 in inhaled air | [73,94] |
| AQI | $AQI = \frac{C_b - C_e}{C_{Pri} - C_e}$ | $C_b$ tracer gas concentration at a point in the breathing zone<br>$C_{pri}$ tracer gas concentration at the primary nozzle exit<br>$C_e$ tracer gas concentration in the exhaust | Evaluate inhaled air quality in the floor diffuser and the secondary nozzle that supplied tracer gas, while primary air was kept free of tracer gas | [39,74,76,85] |
| IF | $IF = \frac{M_{br}}{M_{ge}}$ | $M_{br}$ Particle concentration at breathing level of healthy person, or pollutant mass exposed to person<br>$M_{ge}$ Particle generation concentration, or pollutant mass emitted from a source/polluting person | Represent the proportion of contaminants generated by the infected occupant that is inhaled by the exposed occupant | [77,85,95,103,104,120] |
| DFr | $DFr = \frac{N_{br}}{N_{ge}}$ | $N_{br}$ Number of particles deposited at the vicinity of the exposed person<br>$N_{ge}$ Number of particles generated by the infected person | Assess the cross-infection by the monitoring the particles deposited at the human body of the exposed person and other facilities in the room | [103,104,120] |

**Abbreviations:** personal exposure effectiveness (PEE), pollutant exposure index (PEI), ventilation effectiveness (VE1, VE2, or VE3), intake fraction (IF), deposited fraction (DFr), air quality index (AQI), pollutant exposure reduction (PER).

5.3.2. Thermal Comfort

Regarding thermal comfort involving air temperature, radiant temperature, air velocity, humidity, clothes insulation, and activity, it is recommended to consider all six factors for human response, e.g., by using an integrated thermal comfort model. A recent advanced thermal comfort (ATC) model [196–198] is widely accepted for assessing segmental/overall body thermal comfort and sensation using CFD simulation for PV studies [98,118]. A thermal comfort scale from −4 (very uncomfortable) to +4 (very comfortable), and thermal sensation scale from −4 (very cold) to +4 (very hot) are utilized. This comfort model is more comprehensive and more robust than the previous model [199] for PV studies [37], as it accounts for transient variation of the skin and core temperatures. CFD is more likely coupled with a physiological regulation model through exchanging ambient environment data and manikin segmental skin temperature and heat flux in order to obtain input data to predict thermal sensation and comfort with an ATC model [89,93,99,100,113,119].

The CTM-based equivalent temperature ($t_{eq}$) [200,201] has been widely used to assess the thermal comfort of individual segments for PV studies [39,79,82], and is included in ISO 14505 [202]. It is defined by thermal resistance and sensible heat loss of body segments located in a thermal equilibrium environment, accounting for radiative heat release in some cases [39]. Note that this method does not include the thermoregulation model, and its reliability should be questioned when evaporation from skin is involved, but it has been shown to be in agreement with experimental studies and the ATC model [172]. In addition, the draft risk indicates the local draft discomfort at face exposure to the airflow from a PV nozzle [102,107,108], which has already been adopted in the ASHRAE and ISO thermal comfort standards [203,204]. The airflow interaction in the vicinity of the human body is complex and important, especially when PV nozzles supply fresh air to the breathing zone and penetrate the convective boundary layer around the body, so as to provide more acceptable inhaled air quality without draft complaints [205,206].

To assess the influence of a PV system on thermal comfort, mean skin surface temperature, as an important factor influencing a human's thermal sensation, was employed by calculating Fanger's thermal comfort model parameters and by considering the heat generation and balance for a sleeping person [109,110]. Finally, the commonly used comfort indices—e.g., predicted mean vote (PMV) and predicated percentage of dissatisfaction (PPD), as well as the improvement of overall thermal sensation (DPMV) [107]—were used to examine an occupant's thermal comfort [123]. The impact of PV on CTM thermal comfort is mostly evaluated using the ATC model.

5.3.3. Energy Savings

Significant energy saving can result from maintaining comfort in a wider range by increasing the cooling temperature setpoint or decreasing the heating temperature setpoint [11,48]. CFD can generally not be directly applied to evaluate energy saving, but can provide sufficient information to calculate energy-saving potential compared to traditional mechanical ventilation. For instance, the evaluation criteria are established in terms of the improved temperature setpoint and reduced outdoor airflow rate. The energy-saving potential of PV has been widely investigated by means of energy simulation programs [18,19,50], which are not within the scope of this study. Therefore, only CFD-based valuation indices for energy saving from PV are reviewed in this study.

The direct energy-saving evaluation index mainly accounts for the reduction of cooling loads in the occupied zone using PV devices [96,109,207], or a difference in power consumption of a PV or MV air supply fan due to different flow rates [113]. A cooling load calculation method originally designed for stratified air distribution systems was used [208] by ventilating only the occupied zone and then calculating the cooling energy saving in the coil capacity of DV compared to MV [123]. The coil capacity can be assessed by multiplying the exhaust mass flow rate and the difference between the exhaust temperature and indoor setpoint temperature [111,112]. Similarly, the cooling capacity can be calculated by taking into account the supply flow rate and temperature difference between outside air and supplied air [96,207]. As for PV assisting with a chilled ceiling, the radiant heat load removal

and convective heat load removal should both be considered [119]. The energy consumption for dehumidifying thermal manikins should be considered in addition to the convective cooling load [110].

The reduction in the power consumption of an integrated air supply fan has been assessed in several studies [88,98,117]. Review found that CMPV performs better than conventional MV at maintaining occupants' thermal sensation and inhaled air quality [88–90]. Owing to the advanced design of coaxial nozzles in CMPV, a fan with about 15 W of nominal power supplied fresh air through the ceiling-mounted primary nozzle, and an additional fan with the same power supplied recirculated air through a secondary nozzle, with a varied fresh airflow rate and total supply flow rate. The resulting reduction in power consumption of the supply fan demonstrated the potential energy saving. An evaluation of energy savings with PV used an indirect valuation index called the cooling efficiency (CE) [107]. It is defined as the ratio of additional sensible heat loss removed from the human body to the PRAC's cooling capacity. Note that a portion of a device's cooling capacity is fulfilled by dissipating heat with convective airflow around the human body, resulting in a compromise to achieve high cooling performance and acceptable thermal comfort. In addition, the energy utilization coefficient (EUC), reflecting the difference in average operative temperatures between the occupied and unoccupied zone was used in BTAC with a radiant panel [114]. When EUC is greater than one, it can save energy because less cooling energy is used to remove the heat in the unoccupied zone.

A major finding from reviewing energy saving evaluations for PV systems is that most evaluation indices focus on the terminal device-based energy consumption without considering the effect of the building envelope. Although two articles studied the energy use due to the variation of the envelope heat gain in a bedroom at night [109,110], variations of the envelope heat gain were specified in advance instead of being utilized as an evaluation index in a wider analysis framework. Moreover, the weather data file, equipment schedules, and more detailed control strategies were not considered, resulting in rough evaluations of potential energy savings. Therefore, regardless of whether the direct or indirect method was used, the indices provided a preliminary evaluation of the energy-saving potential of PV systems, and future studies should account for more comprehensive system changes due to the PV presence.

## 6. Conclusions

This paper has reviewed studies of CFD analysis methods for PV systems in indoor built environments. First, this study found that a significant increase in the number of PV system studies was presented using CFD with diverse PV system types associated with different background HVAC systems. Then, besides the basic elements of CFD simulation for PV study, such as turbulence models, boundary condition, computational grids, convergence criteria, and validation of CFD model, this study specifically investigated the modeling of human body and personalized air supply. For human body, this study focused on CTM, deployed in CFD studies on PV to account for its impacts on local thermal state. This study reviewed the influences of CTM body shape, heat exchange over skin surface, and inclusion/exclusion of heat radiation. Overall, the inclusion of radiative heat transfer is fairly important, especially in spaces with radiant floor/panel or external building walls and windows. For personalized air supply, this study reviewed the set up for PV system parameters and evaluation indices for PV performance on inhaled air quality, thermal comfort, and energy-saving. Overall, reviewed studies have demonstrated that CFD is a powerful tool for predicting inhaled air quality, occupants' thermal comfort, and energy-saving potential in buildings. Nevertheless, current studies do not examine interactions of an integrated system that would include thermal comfort of individual occupants, distributed PVs, and other building systems. This integrated approach could uncover major PV system advantages and disadvantages prior to costly implementations in actual buildings. Furthermore, accurate CFD results require careful consideration of the numerical setup, and understanding this is a prerequisite for optimizing PV performance, integration with other building systems, and promoting its application. Through state-of-the-art analyses of current CFD applications in studying PV systems,

future CFD procedures and research interests are suggested to provide a preliminary guideline of PV analyses for building system engineers and researchers.

**Author Contributions:** Conceptualization, J.L. and J.S.; methodology, J.L., S.Z. and J.S.; investigation, J.L. and M.K.K.; supervision, J.S.; writing – original draft, J.L.; writing – review & editing, S.Z., M.K.K. and J.S.; funding acquisition, J.L. and M.K.K.

**Funding:** This research was funded by National Natural Science Foundation of China (51608310) and Science and Technology Plan Project of University in Shandong Province (J16LG07).

**Acknowledgments:** This study is supported by Cluster of Sustainability at the University of Maryland (CITY@UMD). Moreover, Shandong Province Green Building Collaborative Innovation Center Innovation Team Support Program is specially acknowledged.

**Conflicts of Interest:** The authors declared that they have no conflicts of interest to this work.

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
