# Peer review of "A Review of CFD Analysis Methods for Personalized Ventilation (PV) in Indoor Built Environments"

_sustainability, doi:10.3390/su11154166_

Round 1

Reviewer 1 Report

This article has an abundant review of the given research topic and also build up a compact research methodology, but still, have some of the drawbacks need to be proofread. Please have a look the comments below and take these into account to carry out revision.

1.    The nomenclature meaning of CFD and CTM has been mentioned in the abstract. In the paragraph of preface, even the next section, there is no need to appear again, such as lines 66 and 134.

2.     Searching for the term "building environments" via the Internet is generally not added with a word, s. Please, regard it as a reference for revision.

3.    The literature reference number is recommended in order. For the lesser-relevant and cited references, line 87 is used as an example. Both of the 30th and 31st references do not need to appear again.

4.    Please modify the redundant words for line 89 to 90, mainly because of three times showing up the word of "review".

5.    Please, confirm the microenvironment surrounding or surrounding microenvironment, which one is correct.

6.    The description located at line 231 to line 232: The former is a convection-dominated system, i.e., total volume system ventilation [10], including MV (14 studies), DV (14 studies), and UFAD (8 studies) from a total of 32 studies. How many studies are the total number? It is 34 studies, isn't it?

7. Line 90: microenvironment surrounding the human body, which can be said as a term of microclimate. Could you tell readers they're same-otherwise explain their difference?

Author Response

This article has an abundant review of the given research topic and also build up a compact research methodology, but still, have some of the drawbacks need to be proofread. Please have a look the comments below and take these into account to carry out revision.

1. The nomenclature meaning of CFD and CTM has been mentioned in the abstract. In the paragraph of preface, even the next section, there is no need to appear again, such as lines 66 and 134.

Thanks for your comment.

We believe your comment is correct and excellent. However, authors think that for “CFD” and “CTM”, there should be introduced first in the abstract, and then in the preface. We do check all the paper content, there is not additionally redundant description.

Also, we checked online, there are some similar descriptions in the abstract section, and te introduction section.

So, we decided to keep those two nomenclatures, hope this is fine with you. Thank you.

2. Searching for the term "building environments" via the Internet is generally not added with a word, s. Please, regard it as a reference for revision.

Thanks for your comment.

We do checked the current papers, there are some descriptions related to the “building environments”. Here, we follow your suggestions, and change the “building environments” to “indoor built environments”. Hope you are fine with our decision.

This is a highly cited paper: https://www.sciencedirect.com/science/article/pii/S0360132316304334

3. The literature reference number is recommended in order. For the lesser-relevant and cited references, line 87 is used as an example. Both of the 30th and 31st references do not need to appear again.

Thanks for your comment.

I double checked this sentence. Those two references are really important. We insist keeping them. Thanks for your understanding.

4. Please modify the redundant words for line 89 to 90, mainly because of three times showing up the word of "review".

Thanks for your comment.

We have corrected this sentence, and removed one “review” word.

5. Please, confirm the microenvironment surrounding or surrounding microenvironment, which one is correct.

Thanks for your comment.

We double checked those two descriptions. Here, as we would like to mention the environment around the human body, to prevent possible errors, we changed the “surrounding” to “around”, then keep the rest descriptions of “surrounding microenvironment”. Thanks.

6. The description located at line 231 to line 232: The former is a convection-dominated system, i.e., total volume system ventilation [10], including MV (14 studies), DV (14 studies), and UFAD (8 studies) from a total of 32 studies. How many studies are the total number? It is 34 studies, isn't it?

Thanks for your comment.

We actually knew this issue. There are two studies that compares different ventilation systems, therefore, it is 32 studies, not 34 studies. Here, in order to not have any trouble with future authors, we decided to remove the words of “a total of 32 studies”.

7. Line 90: microenvironment surrounding the human body, which can be said as a term of microclimate. Could you tell readers they're same-otherwise explain their difference?

Thanks for your comment.

Yes, you are right. We do think the “microenvironment surrounding the human body” is a term of microclimate. As your comment no. 5 mentioned, we changed the “surrounding” to “around”, and

Reviewer 2 Report

A very good article. The authors have made a very wide range of review of CFD analysis methods for PV in building environments. The article is very useful for the work of scientists in the thematic area. I have no comments.

Author Response

Thanks for your comments. We will do our best to publish this paper.

Reviewer 3 Report

The authors have provided a state-of-the-art CFD analysis of personal ventilation systems. The review is well-written, well-organized, and should be of interest to a wide audience. I provide just a few minor suggestions for improvement below:

- The article is well written, although there are still some grammatical errors throughout that should be corrected, some of which include:

[Line 121] "Totally 60 journal articles"

[125] "even they don't involve PV"

[456] "around human body:"

[855] - "First, review found"

[593, 594] - Sentence on these lines is not written clearly.

- As there are a large number of acronyms in the paper, it is recommended that the authors provide a list of all acronyms at the beginning of the paper"

- Font size in some figures is too small to read

- [347, 348] - there are no units for the surface area values

Author Response

The authors have provided a state-of-the-art CFD analysis of personal ventilation systems. The review is well-written, well-organized, and should be of interest to a wide audience. I provide just a few minor suggestions for improvement below:

- The article is well written, although there are still some grammatical errors throughout that should be corrected, some of which include:

[Line 121] "Totally 60 journal articles"

Thanks for your comment. 

We have made the changes and written as: Total 60 journal articles

[125] "even they don't involve PV"

Thanks for your comment. 

We have made the changes and written as: even they did not involve PV system

[456] "around human body:"

Thanks for your comment. 

We have made the changes and written as: around a human boy, or around the human boy in different sentences.

[855] - "First, review found"

Thanks for your comment. 

We have made the changes and written as:  First, this study found .

[593, 594] - Sentence on these lines is not written clearly.

Thanks for your comment. 

We have made the changes and written as: “The default convergence criteria in CFD programs are not always convincible when the residual root mean square error value is below the criterion (typically 10-4 or 10-5)..

- As there are a large number of acronyms in the paper, it is recommended that the authors provide a list of all acronyms at the beginning of the paper"

Thanks for your comment. 

We have seriously checked the content. The acronyms most came from the tables, because we want to keep concise description. But at the bottom of each table, we provided a list of acronyms for the this table. Please have a look at. So, we decided not provide all the acronyms at the beginning of this paper, hope this is fine with you. Thank you.

- Font size in some figures is too small to read

Thanks for your comment. We have made the changes and used the high quality level pictures (dpi>300). For example, Figure 3

- [347, 348] - there are no units for the surface area values

Thanks for your comment. 

We have made the changes and written as:  1.57 m2 [32, 69], 1.59 m2 [111], or 1.8 m2 [112],.
